# Robustness of Interdependent Networks with Weak Dependency Based on Bond Percolation

**DOI:** 10.3390/e24121801

**Published:** 2022-12-09

**Authors:** Yingjie Qiang, Xueming Liu, Linqiang Pan

**Affiliations:** Key Laboratory of Image Information Processing and Intelligent Control, School of Artificial Intelligence and Automation, Huazhong University of Science and Technology, Wuhan 430074, China

**Keywords:** complex networks, robustness, weak dependency, bond percolation, giant connected component

## Abstract

Real-world systems interact with one another via dependency connectivities. Dependency connectivities make systems less robust because failures may spread iteratively among systems via dependency links. Most previous studies have assumed that two nodes connected by a dependency link are strongly dependent on each other; that is, if one node fails, its dependent partner would also immediately fail. However, in many real scenarios, nodes from different networks may be weakly dependent, and links may fail instead of nodes. How interdependent networks with weak dependency react to link failures remains unknown. In this paper, we build a model of fully interdependent networks with weak dependency and define a parameter α in order to describe the node-coupling strength. If a node fails, its dependent partner has a probability of failing of 1−α. Then, we develop an analytical tool for analyzing the robustness of interdependent networks with weak dependency under link failures, with which we can accurately predict the system robustness when 1−p fractions of links are randomly removed. We find that as the node coupling strength increases, interdependent networks show a discontinuous phase transition when $\alpha <\alpha_{c}$ and a continuous phase transition when $\alpha >\alpha_{c}$. Compared to site percolation with nodes being attacked, the crossover points $\alpha_{c}$ are larger in the bond percolation with links being attacked. This finding can give us some suggestions for designing and protecting systems in which link failures can happen.

## 1. Introduction

Many systems in the real world can be abstracted to nodes, and their relationships can be characterized as links, such as in computer network systems, power systems, transportation systems, etc. [1,2,3]. Network science has provided powerful tools [4,5] that help us understand the universal patterns of complex systems [6,7,8]. The robustness of networks is a key point to study because networks suffer from disturbances all the time [9,10,11]. The initial removal of nodes or links may also cause other nodes to lose their links and disconnect from the giant connected component. Scholars define the area under the robustness curve or the value of the percolation threshold, $p_{c}$, as the measurement of robustness [12,13].

Increasing evidence shows that networks are coupled through dependency connectivities [3] and form interdependent networks or networks of networks [14,15,16]. Interdependency links make a system less robust, since the failures in one network could spread to another one via interdependency links [2,17,18,19]. Buldyrev et al. [10] built a model of fully interdependent networks and found that the interdependent networks showed a discontinuous (first-order) phase transition when a certain fraction of nodes were removed, which was different from the continuous (second-order) phase transitions in single networks [7,20]. A discontinuous phase transition can be used to explain how the Italian blackout in 2003 happened. Since then, plenty of follow-up studies have been carried out to study the robustness of interdependent networks [14,15,21,22,23]. For example, Parshani et al. [23] studied the robustness of partially interdependent networks; Gao et al. [14,15] developed a general framework for analyzing the robustness in networks of networks; Liu et al. [24,25] developed a theoretical tool for analyzing the breakdown phenomena in interdependent directed networks.

As reviewed above, couplings between networks have a great impact on system robustness [26,27,28,29]. Furthermore, different manners in the coupling between networks also bring different behaviors of systems in response to external failures. For example, the inter-similarity [30] or the degree correlations [19,31,32] between coupled networks could improve a system’s robustness. Most of the related studies assume that a dependency link implies a strong dependency relation between nodes, that is, if one node fails, its dependent partner will also immediately fail. However, in reality, networks may be weakly dependent, i.e., a node may survive even if its dependent partner fails in some cases. Liu et al. [33] introduced the concept of weak dependency and defined a parameter α as the node-coupling strength. If one node fails, its dependent partner would have a probability of failing of 1−α. The parameter α can be tuned from zero to one. If α=0, it describes the strongly dependent case, and if α=1, the networks are isolated from one another. Otherwise, the nodes from two networks are weakly dependent.

In addition, there may be different forms of failure. Most of the studies mentioned above focused on site percolation in which failures are injected into nodes. However, in reality, link failures are common, such as in the breakdown of wires of power grids and the cutting off of traffic [17,34,35,36]. The frameworks developed for site percolation cannot be used to describe the bond percolation case, in which the initial failures are link failures. To study bond percolation, Chen. et al. [37] developed an analytical tool for calculating the percolation thresholds of interdependent networks under link failures and found that the percolation threshold $p_{c}$ was smaller in the case of bond percolation compared to that in the case of site percolation. However, how weakly interdependent networks behave in response to link failures remains unknown.

To fill this gap, we built a weakly interdependent network model and developed an analytical tool for studying how the system reacted to initial link failures. By applying the analytical tool to Erdös–Rényi (ER) networks and scale-free (SF) networks, we found that the tool could accurately predict the final giant connected component sizes of the model after randomly removing a certain fraction of the links. Furthermore, we calculated the critical percolation thresholds, $p_{c}$, and the crossover points, $\alpha_{c}$, in order to measure the robustness of the two coupled networks and found that compared to site percolation, the crossover points $\alpha_{c}$ were larger in the case of bond percolation with links being attacked. These results can help in the understanding of system robustness and enable better design of robust systems.

## 2. The Model of Interdependent Networks with Weak Dependency

The model consists of two networks, *A* and *B*, with the same number of nodes *N*. We use the joint degree distribution $P_{A}\left(k\right)$ and $P_{B}\left(k\right)$ to describe their features, where *k* is the degree of a given node and P(k) is the probability of finding that node in the network. For simplicity and without loss of generality, networks *A* and *B* fully depend on each other. All nodes in the two networks depend on a corresponding node according to the no-feedback principle, that is, if node $a_{i}$ in network *A* has a dependency node $b_{j}$ in network *B* and $b_{j}$ has a dependency node $a_{l}$ in network *A*, then l=i.

In the beginning, we remove a fraction of 1−p links in network *A*. If a node loses all of its links, it fails. If a node fails in network *A*, all of the links of its dependency node in network *B* have a probability of failing of 1−α. The nodes that fail in network *B* have the same effect on network *A*. This process continues until the two networks reach stability. A schematic illustration of cascading failure in interdependent networks is shown in Figure 1. Intuitively speaking, the parameter α represents the self-healing capacity of the nodes. α is in the range of 0 and 1. When α is equal to 0, the networks are strongly coupled, and when α is equal to 1, the failures cannot spread, and the two networks are independent. Note that we only remove links from one network to stress the effect of weak dependency, and the two networks’ final states are different. Unlike in the mechanisms of spread that were used previously, the failures propagate from links to links.

Thus, the connection links spread the failures in one network, and the dependency links spread the failures between the networks. The networks break into fragments. As mentioned above, only nodes in the giant connected components (GCCs) function well. After working out the framework, we need to find out the relative size of the GCCs through simulation and calculation. The values of the percolation thresholds are used to describe the robustness of the networks as well.

## 3. Developing the Framework for Analyzing the Robustness in Interdependent Networks with Weak Dependency

The main idea of developing this framework is to construct self-consistent equations based on the definition of variables and their relations. We build the framework on single-layer networks and extend them to double-layer networks by using constraint conditions. The coupling strength parameter, α, is also added, which means that the failed nodes have a probability of surviving of α.

### 3.1. Bond Percolation in Interdependent Networks with Weak Dependency

By using the self-consistent equations [38,39], we can calculate the size of the GCC after an initial random attack and cascading failures. Firstly, we define two probability variables (*x*,$P^{\infty }$) in order to solve the single-layer model. Suppose that there exists a network *A* whose size goes to infinity. Let *x* be the probability that by following an arbitrary direction, a randomly selected link in network *A* leads to the GCC. The degree distribution of *A* can be expressed using the following generating function:(1)$$G_{0}^{A}\left(x\right)=\sum_{k}P_{A}\left(k\right)x^{k}$$
where $P_{A}\left(k\right)$ is the probability of finding a node with *k* links in *A*. The underlying branching process of network A is represented by:(2)$$G_{1}^{A}\left(x\right)=\sum_{k}\frac{P_{A}\left(k\right)\cdot k}{{\langle k\rangle }^{A}}x^{k-1}$$
where ${\langle k\rangle }^{A}$ is the average degree of network *A* and $P_{A}\left(k\right)k/{\langle k\rangle }^{A}$ is the probability of following a link to find a node with degree *k*.

From the above definition, we know that (1−x) is the probability that a randomly selected link does not lead to the GCC. Then, ${\left(1-x\right)}^{k-1}$ is the probability that all of the k−1 links of the node do not lead to the GCC, and $1-{\left(1-x\right)}^{k-1}$ is the probability that at least one link leads to the GCC of that node. Thus, we construct the self-consistent equation of *x*:(3)$$x=\sum_{k}\frac{P_{A}\left(k\right)\cdot k}{{\langle k\rangle }^{A}}\left[1-{\left(1-x\right)}^{k-1}\right]=1-G_{1}^{A}\left(1-x\right)$$

We define $P_{A}^{\infty }$ as the probability that a randomly selected node belongs to the GCC. From the definition, we know that this is also the relative size of the GCC. This means that at least one of its links must lead to the GCC. Thus, we get the following self-consistent equation:(4)$$P_{A}^{\infty }=\sum_{k}P_{A}\left(k\right)\left[1-{\left(1-x\right)}^{k}\right]=1-G_{0}^{A}\left(1-x\right)$$
where ${\left(1-x\right)}^{k}$ means that a node’s *k* links all fail.

An initial link attack means that we need to randomly remove a fraction of 1−p links in network A. Thus, only a fraction of *p* links will remain. Based on Equation (Equation 3), aside from the original probability *x*, the probability of a randomly selected link remaining needs to be multiplied by *p*. For any given *p*, we can solve Equation (Equation 5) to get *x* and substitute it into Equation (Equation 4) to get the final size of the GCC:(5)$$x=\sum_{k}p\cdot \frac{P_{A}\left(k\right)k}{{\langle k\rangle }^{A}}\left[1-{\left(1-x\right)}^{k-1}\right]=p\cdot \left[1-G_{1}^{A}\left(1-x\right)\right]$$

With the fundamentals of the single-layer model, we convert into a fully interdependent two-layer model. $G_{0}^{B}\left(y\right)=\sum_{k}P_{B}\left(k\right)y^{k}$ and $G_{1}^{B}\left(y\right)=\sum_{k}P_{B}\left(k\right)y^{k-1}/{\langle k\rangle }^{B}$ represent the generating function and the underlying branching process of network *B*, respectively. We randomly attack a fraction of 1−p links in network *A*; when the system reaches the steady state, the self-consistent equation is as follows:(6)$$x=p\cdot \left[1-G_{1}^{A}\left(1-x\right)\right]\cdot \left[1-G_{0}^{B}\left(1-y\right)\right]+\alpha \cdot p\cdot \left[1-G_{1}^{A}\left(1-\alpha x\right)\right]\cdot G_{0}^{B}\left(1-y\right)$$

There are two terms on the right-hand side of the equation. We divide the two terms into three parts. Every part is a step in calculating the result of the two-layer model for the first term:*p* is the fraction of links remaining in network *A* after the initial failure.$1-G_{1}^{A}\left(1-x\right)$ is the probability that the end node of the remaining links leads to the GCC of network *A*.$1-G_{0}^{B}\left(1-y\right)$ is the probability that the dependency nodes of the surviving nodes in network *A* of network *B* belong to the GCC of network *B*.

Note that in the steady state, networks *A* and *B* have discrepant sizes because the coupling strength parameter prevents symmetrical initial destruction. In the second term of the right-hand side of Equation (Equation 6), the third part represents the fraction of failing nodes in network *B*, but their dependency nodes in network *A* have the probability α of surviving due to the node-coupling strength parameter. Thus, we need to multiply α to represent the decreasing survival probability of their dependency nodes. Then, we add the two terms to get a self-consistent equation. Similarly to the inference above, we can get the self-consistent equation of *y*; the difference is that the failing of network *B* happens because of the spread of network *A*’s destruction, and the fraction of remaining links is 1, so we omit it in the following equation:(7)$$y=\left[1-G_{1}^{B}\left(1-y\right)\right]\cdot \left[1-G_{0}^{A}\left(1-x\right)\right]+\alpha \cdot \left[1-G_{1}^{B}\left(1-\alpha y\right)\right]\cdot G_{0}^{A}\left(1-x\right)$$

For any given *p*, by substituting the results of *x* and *y* into the following formulas, we can get the relative sizes of networks *A* and *B* in the final state:(8)$$P_{A}^{\infty }=\left[1-G_{0}^{A}\left(1-x\right)\right]\cdot \left[1-G_{0}^{B}\left(1-y\right)\right]+\left[1-G_{0}^{A}\left(1-\alpha x\right)\right]\cdot G_{0}^{B}\left(1-y\right)$$
(9)$$P_{B}^{\infty }=\left[1-G_{0}^{B}\left(1-y\right)\right]\cdot \left[1-G_{0}^{A}\left(1-x\right)\right]+\left[1-G_{0}^{B}\left(1-\alpha y\right)\right]\cdot G_{0}^{A}\left(1-x\right)$$

Because we did not remove nodes, the initial survival probability of the nodes is 1. The surviving nodes are divided into two parts: The dependency nodes of one survived and the other’s dependency nodes failed. For the first part, they are strictly limited by the two networks’ survival probabilities. The nodes in the second part survived because of the coupling strength parameter α, although their dependency nodes failed. The survival probability of the nodes in network *A* or *B* is added to the two terms.

### 3.2. Applying the Framework to Interdependent ER Networks

We apply our framework to two-layer ER networks whose degree distributions follow the Poissonian distribution: $P_{A}\left(k\right)=\frac{e^{-\langle k_{A}\rangle }{\langle k_{A}\rangle }^{k}}{k!}$ and $P_{B}\left(k\right)=\frac{e^{-\langle k_{B}\rangle }{\langle k_{B}\rangle }^{k}}{k!}$. For simplicity and without loss of generality, we assume that $\langle k_{A}\rangle =\langle k_{B}\rangle$. By substituting the degree distribution into (Equation 6) and (Equation 7), we obtain:(10)$$x=p\cdot [\left(1-e^{-\langle k_{A}\rangle x}\right)\cdot \left(1-e^{-\langle k_{B}\rangle y}\right)+\alpha \cdot \left(1-e^{-\langle k_{A}\rangle \alpha x}\right)\cdot e^{-\langle k_{B}\rangle y}]$$
(11)$$y=\left(1-e^{-\langle k_{B}\rangle y}\right)\cdot \left(1-e^{-\langle k_{A}\rangle x}\right)+\alpha \cdot \left(1-e^{-\langle k_{B}\rangle \alpha y}\right)\cdot e^{-\langle k_{A}\rangle x}$$

For any given *p* between 0 and 1, after getting the values of *x* and *y* with the two formulas in Equations (Equation 10) and (Equation 11), we substitute them into the following formulas to get the final results:(12)$$P_{A}^{\infty }=\left(1-e^{-\langle k_{A}\rangle x}\right)\cdot \left(1-e^{-\langle k_{B}\rangle y}\right)+\left(1-e^{-\langle k_{A}\rangle \alpha x}\right)\cdot e^{-\langle k_{B}\rangle y}$$
(13)$$P_{B}^{\infty }=\left(1-e^{-\langle k_{B}\rangle y}\right)\cdot \left(1-e^{-\langle k_{A}\rangle x}\right)+\left(1-e^{-\langle k_{B}\rangle \alpha y}\right)\cdot e^{-\langle k_{A}\rangle x}$$

The theoretical results are shown by the solid lines in Figure 2. We also obtained experimental results through a simulation and present them as symbols. This shows that the theoretical predictions were accurate. With the growth of α, the system became more robust. On the one hand, the area under the percolation curves became larger. On the other hand, the critical percolation transition points $p_{c}$ became smaller. Furthermore, we observed that there was a critical point $\alpha_{c}$ at which the transition form changed from a first-order transition to a second-order transition. The size of network *A*’s GCC jumped from 0 to a finite value when α=0.5, but it changed continuously to a negative value when α=0.6. We could observe the phenomenon more directly in network *B*, since when $\alpha >\alpha_{c}$, the size of network *B*’s GCC was always a negative value.

### 3.3. Applying the Framework to Interdependent SF Networks

Another application was to a scale-free (SF) network; the normalized degree distributions were $P_{A}\left(k\right)=P_{B}\left(k\right)=\frac{\sum_{k_{min}}^{k_{max}}\left[{\left(k+1\right)}^{\left(1-\lambda \right)}-k^{\left(1-\lambda \right)}\right]}{{\left(k_{max}+1\right)}^{1-\lambda }-k_{min}^{1-\lambda }}$, where $k_{min}=2$, $k_{max}=2000$, and the power law of these networks was λ=2.7. The theoretical results and experimental results also coincided well, as shown in Figure 3.

By substituting the degree distribution into Equations (Equation 6) and (Equation 7), we obtain:(14)$$\begin{array}{c} x=p\cdot [\left(1-\frac{\sum_{k_{min}}^{k_{max}}\left[{\left(k+1\right)}^{\left(1-\lambda \right)}-k^{\left(1-\lambda \right)}\right]\cdot {\left(1-x\right)}^{\left(k-1\right)}}{\sum_{k_{min}}^{k_{max}}\left[{\left(k_{max}+1\right)}^{1-\lambda }-k_{min}^{1-\lambda }\right]k}\right)\cdot \left(1-\frac{\sum_{k_{min}}^{k_{max}}\left[{\left(k+1\right)}^{\left(1-\lambda \right)}-k^{\left(1-\lambda \right)}\right]\cdot {\left(1-y\right)}^{k}}{{\left(k_{max}+1\right)}^{1-\lambda }-k_{min}^{1-\lambda }}\right)\\ +\alpha \cdot \left(1-\frac{\sum_{k_{min}}^{k_{max}}\left[{\left(k+1\right)}^{\left(1-\lambda \right)}-k^{\left(1-\lambda \right)}\right]\cdot {\left(1-\alpha x\right)}^{\left(k-1\right)}}{\sum_{k_{min}}^{k_{max}}\left[{\left(k_{max}+1\right)}^{1-\lambda }-k_{min}^{1-\lambda }\right]k}\right)\cdot \frac{\sum_{k_{min}}^{k_{max}}\left[{\left(k+1\right)}^{\left(1-\lambda \right)}-k^{\left(1-\lambda \right)}\right]\cdot {\left(1-y\right)}^{k}}{{\left(k_{max}+1\right)}^{1-\lambda }-k_{min}^{1-\lambda }}] \end{array}$$
(15)$$\begin{array}{c} y=\left(1-\frac{\sum_{k_{min}}^{k_{max}}\left[{\left(k+1\right)}^{\left(1-\lambda \right)}-k^{\left(1-\lambda \right)}\right]\cdot {\left(1-y\right)}^{\left(k-1\right)}}{\sum_{k_{min}}^{k_{max}}\left[{\left(k_{max}+1\right)}^{1-\lambda }-k_{min}^{1-\lambda }\right]k}\right)\cdot \left(1-\frac{\sum_{k_{min}}^{k_{max}}\left[{\left(k+1\right)}^{\left(1-\lambda \right)}-k^{\left(1-\lambda \right)}\right]\cdot {\left(1-x\right)}^{k}}{{\left(k_{max}+1\right)}^{1-\lambda }-k_{min}^{1-\lambda }}\right)\\ +\alpha \cdot \left(1-\frac{\sum_{k_{min}}^{k_{max}}\left[{\left(k+1\right)}^{\left(1-\lambda \right)}-k^{\left(1-\lambda \right)}\right]\cdot {\left(1-\alpha y\right)}^{\left(k-1\right)}}{\sum_{k_{min}}^{k_{max}}\left[{\left(k_{max}+1\right)}^{1-\lambda }-k_{min}^{1-\lambda }\right]k}\right)\cdot \frac{\sum_{k_{min}}^{k_{max}}\left[{\left(k+1\right)}^{\left(1-\lambda \right)}-k^{\left(1-\lambda \right)}\right]\cdot {\left(1-x\right)}^{k}}{{\left(k_{max}+1\right)}^{1-\lambda }-k_{min}^{1-\lambda }} \end{array}$$

For any given *p* between 0 and 1, after getting the values of *x* and *y* with the two formulas in Equations (Equation 14) and (Equation 15), we substitute them into the following formulas to get the final results:(16)$$\begin{array}{c} P_{A}^{\infty }=\left(1-\frac{\sum_{k_{min}}^{k_{max}}\left[{\left(k+1\right)}^{\left(1-\lambda \right)}-k^{\left(1-\lambda \right)}\right]\cdot {\left(1-x\right)}^{k}}{{\left(k_{max}+1\right)}^{1-\lambda }-k_{min}^{1-\lambda }}\right)\cdot \left(1-\frac{\sum_{k_{min}}^{k_{max}}\left[{\left(k+1\right)}^{\left(1-\lambda \right)}-k^{\left(1-\lambda \right)}\right]\cdot {\left(1-y\right)}^{k}}{{\left(k_{max}+1\right)}^{1-\lambda }-k_{min}^{1-\lambda }}\right)\\ +\left(1-\frac{\sum_{k_{min}}^{k_{max}}\left[{\left(k+1\right)}^{\left(1-\lambda \right)}-k^{\left(1-\lambda \right)}\right]\cdot {\left(1-\alpha x\right)}^{k}}{{\left(k_{max}+1\right)}^{1-\lambda }-k_{min}^{1-\lambda }}\right)\cdot \frac{\sum_{k_{min}}^{k_{max}}\left[{\left(k+1\right)}^{\left(1-\lambda \right)}-k^{\left(1-\lambda \right)}\right]\cdot {\left(1-y\right)}^{k}}{{\left(k_{max}+1\right)}^{1-\lambda }-k_{min}^{1-\lambda }} \end{array}$$
(17)$$\begin{array}{c} P_{B}^{\infty }=\left(1-\frac{\sum_{k_{min}}^{k_{max}}\left[{\left(k+1\right)}^{\left(1-\lambda \right)}-k^{\left(1-\lambda \right)}\right]\cdot {\left(1-y\right)}^{k}}{{\left(k_{max}+1\right)}^{1-\lambda }-k_{min}^{1-\lambda }}\right)\cdot \left(1-\frac{\sum_{k_{min}}^{k_{max}}\left[{\left(k+1\right)}^{\left(1-\lambda \right)}-k^{\left(1-\lambda \right)}\right]\cdot {\left(1-x\right)}^{k}}{{\left(k_{max}+1\right)}^{1-\lambda }-k_{min}^{1-\lambda }}\right)\\ +\left(1-\frac{\sum_{k_{min}}^{k_{max}}\left[{\left(k+1\right)}^{\left(1-\lambda \right)}-k^{\left(1-\lambda \right)}\right]\cdot {\left(1-\alpha y\right)}^{k}}{{\left(k_{max}+1\right)}^{1-\lambda }-k_{min}^{1-\lambda }}\right)\cdot \frac{\sum_{k_{min}}^{k_{max}}\left[{\left(k+1\right)}^{\left(1-\lambda \right)}-k^{\left(1-\lambda \right)}\right]\cdot {\left(1-x\right)}^{k}}{{\left(k_{max}+1\right)}^{1-\lambda }-k_{min}^{1-\lambda }} \end{array}$$

The theoretical results are shown by the solid lines in Figure 3. We also obtained experimental results through a simulation and present them as symbols. This shows that the theoretical predictions were also accurate.

## 4. The Crossover Points of Phase Transitions

In the previous section, we measured the robustness through percolation theory. We obtained *x* and *y* through a dichotomy and substituted them into Equations (Equation 12) and (Equation 13) to get $P_{A}^{\infty }$ and $P_{B}^{\infty }$. As shown in Figure 2 and Figure 3, when the remaining fraction of links crossed over a value $p_{c}$, a GCC emerged. In this section, we first calculate the value of the percolation thresholds $p_{c}$. Then, we calculate the crossover points that divide the first-order phase transition and the second-order phase transition.

### 4.1. The Percolation Thresholds

We calculated $p_{c}$ for different values α by judging the sign of $P_{A}^{\infty }$ and demonstrate the results in Figure 4. In panels (a) and (b) of the figure, the $p_{c}$ values for bond percolation are represented by solid lines, while the $p_{c}$ values for site percolation are represented by dashed lines. The percolation thresholds for site percolation are larger, and we assume that the robustness of bond percolation is better. We also validated our predictions by comparing the robustness curves. In panels (c) and (d) of the figure, the area under the robustness curves of bond percolation is larger with the same α, which means that the networks are more robust to bond failures.

### 4.2. The Crossover Points

In order to find the critical point $\alpha_{c}$, we observe the graphical solutions of Equations (Equation 10) and (Equation 11). Equation (Equation 11) is not affected by *p*, since we do not initially destroy network *B*. In Figure 5, we present Equations (Equation 10) and (Equation 11) on the x,y plane and set α=0.3 and $\langle k_{A}\rangle =\langle k_{B}\rangle =4$. If the two curves have an intersection aside from the trivial solution x=y=0, the positive solutions make $P_{A}^{\infty }>0$ and $P_{B}^{\infty }>0$. By adjusting the parameter *p*, Equation (Equation 10) moves from the left to the right of the plane. When *p* is equal to $p_{c}=0.3919$, the two curves have a tangent point, which means that the value of $P_{A}^{\infty }$ jumps from 0 to a nonzero value and the phase transition is discontinuous. However, when α=0.7, the model undergoes a continuous phase transition, and the solution is unique when *p* is given. With decreasing *p*, the solutions of Equation (Equation 10) decrease continuously to 0 and $\frac{dx}{dy}\cdot \frac{dy}{dx}\neq 1$ when $p=p_{c}$ because the two lines cannot have a tangent. The differences between the two conditions are shown in Figure 5. Thus, we can decide on the kind of phase transition by finding the critical point $p_{c}$ and obtaining the relevant solutions of x,y to verify the whether $\frac{dx}{dy}\cdot \frac{dy}{dx}=1$:(18)$$\frac{dx}{dy}=\frac{pG_{0}^{B^{\prime }}\left(1-y\right)\cdot \left[1-\alpha -G_{1}^{A}\left(1-x\right)+\alpha G_{1}^{A}\left(1-\alpha x\right)\right]}{1-pG_{1}^{A^{\prime }}\left(1-x\right)\left[1-G_{0}^{B}\left(1-y\right)\right]-\alpha^{2}pG_{1}^{A^{\prime }}\left(1-\alpha x\right)G_{0}^{B}\left(1-y\right)}$$
(19)$$\frac{dy}{dx}=\frac{G_{0}^{A^{\prime }}\left(1-x\right)\cdot \left[1-\alpha -G_{1}^{B}\left(1-y\right)+\alpha G_{1}^{B}\left(1-\alpha y\right)\right]}{1-G_{1}^{B^{\prime }}\left(1-y\right)\left[1-G_{0}^{A}\left(1-x\right)\right]-\alpha^{2}G_{1}^{B^{\prime }}\left(1-\alpha y\right)G_{0}^{A}\left(1-x\right)}$$

We took 20 values evenly spaced between 0 and 1 to find the intervals of $\alpha_{c}$, whose left endpoint was a first-order transition and whose right endpoint was a second-order transition. Through dichotomy, we found the values of $\alpha_{c}$ of ER networks with different average degrees 〈k〉 and SF networks with different minimum degrees $k_{min}$. We marked the results on the corresponding lines. With the increase in α, $p_{c}$ decreased from the value of a strong coupling model to a single-layer model. When α was close to $\alpha_{c}$, the changes in $p_{c}$ were more drastic. After going across $\alpha_{c}$, $p_{c}^{II}$ gently decreased. Networks that had larger percolation thresholds also had larger crossover points, as shown in Figure 4. However, the crossover points for site percolation were smaller than those for bond percolation, while the percolation thresholds were larger, as shown in Table 1. Thus, the crossover points cannot become the measurement of robustness. Moreover, this phenomenon suggests that link failure is more likely to lead to collapses of the networks even though they are more robust than those in site percolation. Because removing links seems to reserve nodes in networks, the average degree actually decreases. This makes the networks fragile, and a small disturbance can thoroughly destroy the system.

## 5. Conclusion

In this paper, we applied a weakly coupled mechanism to interdependent networks based on bond percolation. In the framework that we constructed, sites only played the role of a medium. The coupling strength parameter α controlled the spreading strength of the dependency nodes’ failure. When $\alpha <\alpha_{c}$, the coupling strength was strong, and with the increase in α, $p_{c}$ decreased faster than when $\alpha >\alpha_{c}$ with the growth of α. That is to say, after going through the crossover point $\alpha_{c}$, the system was always robust. Interestingly, we know that the sizes of the GCCs were bigger and the values of the percolation thresholds were smaller for bond percolation than those for site percolation. For the crossover points, $\alpha_{c}$ was greater, which means that the case in bond percolation was easier to break down. This is because when we remove a fraction of the links, the average degree decreases, and the networks become less robust. Thus, we need to improve the self-protection ability for cases in which interdependent networks’ edges are continuously attacked, such as in the breakdown of business relationships. Moreover, since attacks on links and nodes lead to different results, we need to make an effort to study the relationship between links and nodes.

## Figures and Tables

**Figure 1 entropy-24-01801-f001:**
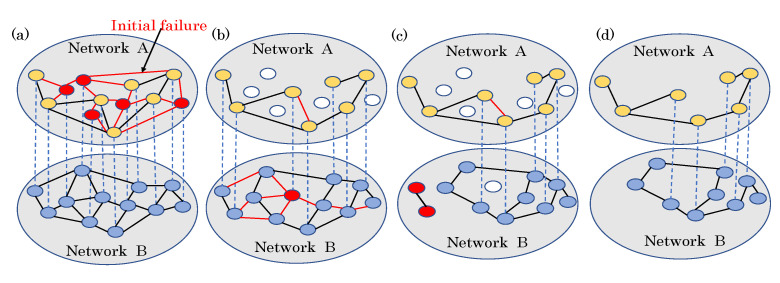
Schematic illustration of cascading failure in interdependent networks. Network A and Network B are fully dependent, as shown by the dependency links, which are represented as blue lines in the figure. The nodes in both networks are initially functional. (**a**) Initial failures occur in Network A. The red edges are failed edges. The red nodes failed because all of their links were removed. (**b**) The failed nodes were removed, while the nodes’ links that depended on the failed nodes of another network had a probability of failing of 1−α. (**c**) Nodes that were not located in the giant components lost function. (**d**) The network eventually reached a stable state, and the remaining nodes were all located in the giant connected components.

**Figure 2 entropy-24-01801-f002:**
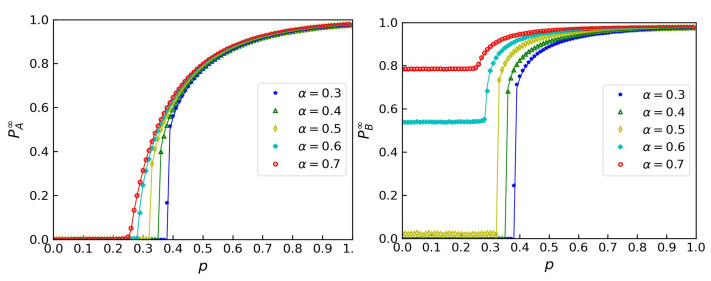
The sizes of the giant components $P_{A}^{\infty }$ (**left**) and $P_{B}^{\infty }$ (**right**) in coupled ER networks with $\langle k_{A}\rangle =\langle k_{B}\rangle =4$ when a fraction of 1−p of the links were randomly removed from network A. The solid lines represent the theoretical predictions. The symbols represent simulation results from 40 iterations on networks with $10^{5}$ nodes.

**Figure 3 entropy-24-01801-f003:**
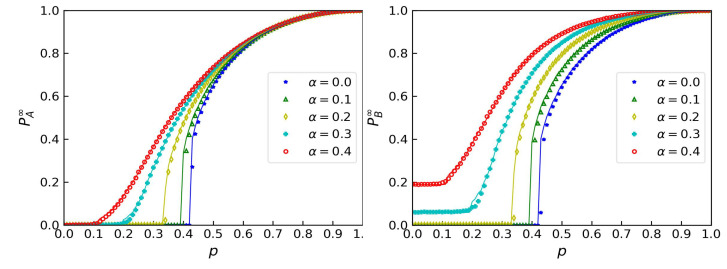
The sizes of the giant components $P_{A}^{\infty }$ (**left**) and $P_{B}^{\infty }$ (**right**) in coupled SF networks with $k_{min}=2,k_{max}=2000$ and λ=2.7 when a fraction of 1−p of the links were randomly removed from network A. The solid lines represent the theoretical predictions. The symbols represent the simulation results from 40 iterations on networks with $10^{5}$ nodes.

**Figure 4 entropy-24-01801-f004:**
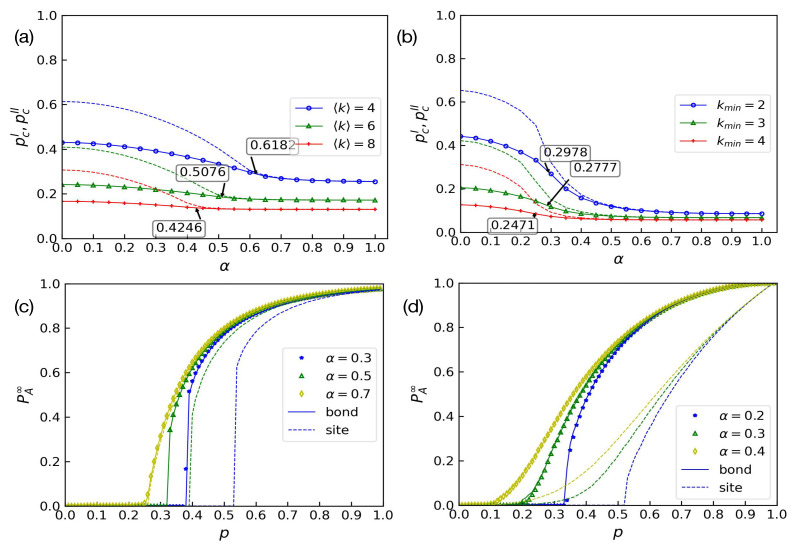
A comparison of bond percolation and site percolation. (**a**) Percolation thresholds of ER networks. The dashed lines represent site percolation, while the solid lines represent bond percolation. Different colors of lines represent different average degrees 〈k〉 of the ER networks. (**b**) Percolation thresholds of SF networks, the power law of which is λ=2.7 and the maximum values of degrees of which are $k_{max}=316$. Different colors of the different lines represent different minimum degrees of the SF networks. The crossover points for bond percolation are marked on the corresponding curves. (**c**) Sizes of giant connected components after a fraction of 1−p sites or bonds are removed in ER networks. The solid lines represent bond percolation and the dashed lines represent site percolation. (**d**) Sizes of giant connected components after a fraction of 1−p sites or bonds are removed in SF networks.

**Figure 5 entropy-24-01801-f005:**
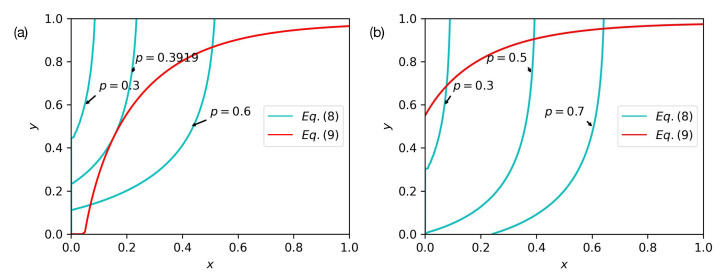
Bond percolation in interdependent networks with weak dependency. The red line represents Equation (Equation 11), and it is not affected by *p* when α is a constant value. The blue lines represent Equation (Equation 10). (**a**) As *p* increases from 0 to 1 when $\alpha <\alpha_{c}$, the two curves have a tangent point. (**b**) However, when $\alpha >\alpha_{c}$, the solutions of the two equations continuously decrease to 0.

**Table 1 entropy-24-01801-t001:** Values of $\alpha_{c}$ for ER networks and SF networks.

	ER Networks			SF Networks		
	**〈*k*〉 = 4**	**〈*k*〉 = 6**	**〈*k*〉 = 8**	***k_min_* = 2**	***k_min_* = 3**	***k_min_* = 4**
bond percolation	0.6182	0.5076	0.4246	0.2978	0.2777	0.2471
site percolation	0.5737	0.4721	0.4056	0.2913	0.2536	0.2228

## Data Availability

Not applicable.

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
