# Peer review of "Robustness of Interdependent Networks with Weak Dependency Based on Bond Percolation"

_entropy, 2022, doi:10.3390/e24121801_

Round 1

Reviewer 1 Report

This manuscript studies the robustness of weaklydependent networks concerning link failure cascades. The authors develop an analytical tool to predict the robustness of dependent networks when 1-p fractions of links are randomly removed. By applying the theory to interdependent ER networks and SF networks, they find that interdependent networks with weak dependencies are more robust in responses to random link failures. They found that the interdependent networks first show a discontinuous phase transition and then a continuous phase transition, as the node coupling strength increases across a crossover point. By comparing to the cases of initial node failures, the interdependent networks are more robust to initial link failures. These findings provide some suggestions for protecting and designing the dependent networks.

The manuscript presents a nice work with both a clear analytical framework and important results. I have some minor comments, which are suggested to be revised:

1. In line 173, the description of “the giant components” is not clear. 2. Please revise the English of the third section, as some conceptions are provided without clear explanation, which makes the paper not easy to understand, such as the definition of generating functions and the underlying branching process. 3. The fourth part is about the percolation thresholds and the crossover points, which are two different concepts. I suggest dividing them separately.

Reviewer 2 Report

This manuscript presents a framework for measuring the robustness of weakly dependent complex networks based on bond percolation, generating functions and self-consistent equations. Then the authors apply the framework to interdependent ER and SF networks. By measuring the area under percolation curves and percolation thresholds, which characterizes network robustness, they conclude that the weak coupling parameter can improve the robustness of dependent networks under link failures. They also calculate the crossover points which divide the first-order percolation and the second-order percolation.

I think it is an interesting work but the analysis is insufficient because the authors should analyze the relationship between “crossover points” and the “percolation thresholds” in bond percolation. The manuscript is written clearly but still has some mistakes. According to the topic's significance, I recommend its publication on Entropy. I have four suggestions for it:

1.  “The crossover points of bond percolation are larger than site percolation, but the percolation thresholds are smaller.” This comparison lacks significance. And in this work, only for bond percolation, networks with different parameters have larger percolation thresholds and also have larger crossover points, how to explain it?

2.     In line 122, coming up with the abbreviation of “GC”, I wonder if it is a mistake or a new definition.

3.     Line 132 describes “a steady state”, I think “the final state” rather than “a steady state” is more in line with the article.

4.     In line 149, “the GCC of the networks becomes bigger relatively with the same p.” The authors have defined the measurement of robustness before. I suggest that the authors reference the measurement in the introduction.

Reviewer 3 Report

In this paper, the robustness of interdependent networks with weak dependency under link failures is considered. Compared to site percolation where nodes are attacked, the crossover points obtained here are larger in the bond percolation. It is an interesting result.

Some comments are: 

1. The English expression should be polished.

2. There are some grammar errors in some sentences, e.g., "Increasing evidence shows that networks coupled through dependency connectivities[3], forming interdependent networks or networks of networks [14–16]."

3. In line 78, page 3, "if node a_i in network A has a dependency node b_j in network B and b_j has a dependency node ak in network A, then k = i".  Note that k has been defined as degree of a node, and so what is the meaning of i?
